# Triple-Emulsion-Based Antibubbles: A Step Forward in Fabricating Novel Multi-Drug Delivery Systems

**DOI:** 10.3390/pharmaceutics15122757

**Published:** 2023-12-12

**Authors:** Rabia Zia, Albert T. Poortinga, Akmal Nazir, Salahdein Aburuz, Cornelus F. van Nostrum

**Affiliations:** 1Department of Pharmaceutics, Utrecht Institute for Pharmaceutical Sciences, Utrecht University, 3584 CG Utrecht, The Netherlands; r.zia@uu.nl; 2Department of Mechanical Engineering, Polymer Technology, Eindhoven University of Technology, 5612 AZ Eindhoven, The Netherlands; albert.poortinga@bether-encapsulates.nl; 3Department of Food Science, College of Agriculture and Veterinary Medicine, United Arab Emirates University, Al Ain P.O. Box 15551, United Arab Emirates; akmal.nazir@uaeu.ac.ae; 4Department of Pharmacology & Therapeutics, College of Medicine and Health Sciences, United Arab Emirates University, Al Ain P.O. Box 15551, United Arab Emirates; saburuz@uaeu.ac.ae

**Keywords:** triple emulsion, Pickering emulsion, nano-emulsion, proteins, antibubbles, drug delivery

## Abstract

Developing carriers capable of efficiently transporting both hydrophilic and lipophilic payloads is a captivating focus within the pharmaceutical and drug delivery research domain. Antibubbles, constituting an innovative encapsulation system designed for drug delivery purposes, have garnered scientific interest thanks to their distinctive water-in-air-in-water (W_1_/A/W_2_) structure. However, in contrast to their precursor, i.e., nanoparticle-stabilized W_1_/O/W_2_ double emulsion, traditional antibubbles lack the ability to accommodate a lipophilic payload, as the intermediary (volatile) oil layer of the emulsion is replaced by air during the antibubble fabrication process. Therefore, here, we report the fabrication of triple-emulsion-based antibubbles (O_1_/W_1_/A/W_2_), in which the inner aqueous phase was loaded with a nanoemulsion stabilized by various proteins, including whey, soy, or pea protein isolates. As model drugs, we employed the dyes Nile red in the oil phase and methylene blue in the aqueous phase. The produced antibubbles were characterized regarding their size distribution, entrapment efficiency, and stability. The produced antibubbles demonstrated substantial entrapment efficiencies for both lipophilic (ranging from 80% to 90%) and hydrophilic (ranging from 70% to 82%) components while also exhibiting an appreciable degree of stability during an extended rehydration period of two weeks. The observed variations among different antibubble variants were primarily attributed to differences in protein concentration rather than the type of protein used.

## 1. Introduction

Antibubbles, constituting a newly developed encapsulate for drug delivery applications, have gained scientific attention due to their unique structure, that is, a gaseous layer surrounding inner aqueous droplets (i.e., water-in-air-in-water, W_1_/A/W_2_) [1]. The gaseous layer protects the drug-containing inner droplets by minimizing interaction with the external environment. However, the water–air interfaces must be sufficiently stable to allow antibubbles to be used as drug delivery systems. Nanoparticle-stabilized antibubbles offer enhanced stability and are preferred over surfactant-based antibubbles. They are produced when a Pickering double emulsion (W_1_/O/W_2_) is freeze-dried and then rehydrated [2]. The middle layer of the double emulsion is always a volatile oil that sublimates along with the inner and outer water phases during freeze-drying. However, upon rehydration, the inner and outer water phases regain water, whereas the middle air layer (previously an oil layer) remains intact. To ensure the preservation of the structure during lyophilization, it is essential to introduce a cryoprotective element (often a carbohydrate) into both the inner and outer water phases, thereby preventing structural collapse. To attain a complete delivery of the drug from the cores of the encapsulates, the nanoparticles at both the W/A and A/W interfaces must be dislocated. This will subsequently release the drug from the inner cores into the surrounding environment. The nature of potential triggers for drug release can vary based on the type of nanoparticles stabilizing the interface and the specific location where drug release is needed. Antibubbles are unique in the sense that drug release can be initiated through the displacement of interfacially adsorbed nanoparticles by site-specific triggers. For instance, gastric delivery can be obtained via the acid-dependent dissolution of the interfacial particles [2]. Several other mechanisms can be employed as a specific trigger (e.g., ultrasounds [3], temperature [4], or bile salts [5]) at a specific site (e.g., mouth, stomach, intestine, or colon), causing the nanoparticles at the interface to dissolve or dislocate, releasing the gas shell and ultimately the inner drug-containing cores.

Up till now, only double-emulsion templated antibubbles have been prepared (e.g., described in the above paragraph), which are effective for encapsulating water-soluble drugs. However, in many cases, drugs are lipophilic, or a lipophilic drug must be delivered together with a hydrophilic drug. Hence, there is a necessity to develop antibubbles capable of encapsulating lipophilic substances as well. In this context, we introduce novel triple-emulsion templated antibubbles (O/W_1_/A/W_2_), enabling the simultaneous encapsulation of both hydrophilic and lipophilic drugs. The proof-of-concept formation and stability of this novel structure were tested through the loading of lipophilic and hydrophilic dyes (i.e., Nile red and methylene blue, respectively) in a primary oil-in-water (O_1_/W_1_) nano-emulsion. The rest of the procedure was similar to the preparation of double-emulsion templated antibubbles as reported previously [5], i.e., the primary O_1_/W_1_ emulsion was used to create the O_1_/W_1_/O_2_ (double) and O_1_/W_1_/O_2_/W_2_ (triple) emulsions. The latter was freeze-dried, and then rehydrated to get O_1_/W_1_/A/W_2_ antibubbles.

A crucial factor when creating such antibubbles is employing a suitable stabilization mechanism for the primary O_1_/W_1_ emulsion. Principally, a surfactant capable of generating a nano-sized emulsion while having minimal interactions with other interfaces within the antibubble structure (usually stabilized by silica nanoparticles) is a viable choice to contemplate. Low-molecular-weight surfactants (like Tween 20) are fundamentally unsuitable due to their widely acknowledged ability to modify the wettability of solid particles, leading to their displacement from the interface [6]. Conversely, employing silica particles in W_1_ (similar to W_2_) is also unfavorable, given the challenges in achieving a nano-emulsion. Therefore, we opted to use proteins to stabilize the O_1_/W_1_ primary emulsion given their established ability to generate stable nano-emulsion systems [7]. Whey protein, a highly utilized animal protein, has sparked concerns regarding its allergenic potential, prompting a shift towards more user-friendly plant-based proteins. This transition is reinforced by the sustainability benefits associated with plant proteins, as well as their compatibility with vegetarian, vegan, and diverse dietary preferences dictated by culture and religion [8]. Consequently, the current study investigates the stabilization of the primary emulsion using whey protein as well as two plant proteins, namely soy and pea. The entrapment of hydrophilic and lipophilic components in different antibubble variants, and the overall integrity of the antibubble structure (during an extended rehydration period), were assessed to identify the optimal choice for stabilizing triple-emulsion-based antibubbles.

## 2. Materials and Methods

### 2.1. Materials

The hydrophobized fumed silica particles, AEROSIL^®^ R972 and AEROSIL^®^ R816, were kindly provided by Evonik (Dubai, United Arab Emirates). The AEROSIL^®^ R972 silica particles, being the most hydrophobic, were employed to stabilize the W/O emulsion. The less hydrophobic AEROSIL^®^ R816 silica particles, which can easily be dispersed in the water phase, were used to stabilize the O/W emulsion. Both types of silica particles were utilized in emulsification without any pre-treatment. The median diameter of both types of silica particles was typically around 200 nm [2]. Whey protein isolate (WPI, 92% purity), soy protein isolate (SPI, 90% purity), and pea protein isolate (PPI, 80% purity) were supplied by Myprotein (Manchester, UK). Cyclohexane (≥99.5%) was purchased from Honeywell (Charlotte, NC, USA). Medium-chain triglyceride (MCT) oil was provided by NOW^®^ Sports (Bloomingdale, IL, United States). Maltodextrin (dextrose equivalent: 16.5–19.5), methylene blue (≥95%), Nile red, and Tween^®^ 20 were provided by Sigma Aldrich (St. Louis, MO, USA).

### 2.2. Formation of Silica- or Protein-Stabilized O_1_/W_1_ Primary Emulsions

O_1_/W_1_ primary emulsions were fabricated using R816 silica nanoparticles (SNP) or proteins (WPI, SPI, or PPI). Briefly, 0.5% of SNP and 10% of maltodextrin were added to 15 mL of water and sonicated (Branson Digital Sonifier, SFX 550, Emerson, Brookfield, CT, USA) at 50% power for 1 min to disperse the nanoparticles in the aqueous phase. Then, MCT oil (containing 0.1% Nile red) was added to this mixture at a concentration of 5% of the aqueous phase. The coarse emulsion was prepared using T25 digital ULTRA-TURRAX^®^ (IKA, Staufen, Germany) at 10,000 rpm for 1 min. Then, the coarse emulsion was subjected to a high-pressure homogenizer (APV 1000, APV SYSTEMS, Copenhagen, Denmark) at a pressure of 500 bar for three homogenization cycles. For the preparation of protein stabilized primary emulsions, each protein was solubilized in distilled water at a concentration of 2% (based on their purity levels). The protein solutions were stirred using a magnetic stirrer for 1 h at room temperature. For proper hydration, the protein solutions were kept in a refrigerator overnight at 4 °C. This was followed by centrifugation at 10,000 rpm at 4 °C for 30 min to remove any undissolved or suspended material from the protein mixtures. The supernatants were carefully collected from the top, leaving behind the sediments. Afterwards, MCT oil (containing 0.1% Nile red) was added to each protein suspension at a concentration of 5%. Similar to SNP-based emulsions, coarse emulsions were made using ULTRA-TURRAX^®^ at 10,000 rpm for 1 min. The coarse emulsions were then subjected to high-pressure homogenization at 500 bar of pressure for three cycles to produce protein-stabilized nano-emulsions.

### 2.3. Determination of Drop Size Distribution of Primary Emulsions

The primary emulsions were characterized with respect to their droplet size distribution using a laser diffraction particle size analyzer (Malvern Mastersizer 3000, Malvern Panalytical Ltd., Malvern, UK). Refractive indices of 1.46 and 1.33 were used for MCT oil and aqueous phases, respectively. The emulsion samples were gradually added into the wet sample dispersion unit (Hydro EV) using a 3 mL bubble pipette until 10% obscuration level was reached. Each measurement cycle was run three times to record the droplet size distribution, and then the average median diameter and span value (a measure of droplet uniformity) were obtained.

### 2.4. Formation of O_1_/W_1_/O_2_ Double Emulsion

Protein-stabilized O_1_/W_1_ primary emulsions were further used to prepare O_1_/W_1_/O_2_ double-emulsion variants. Briefly, 5 mL of an O_1_/W_1_ nano-emulsion was acquired in a 15 mL plastic tube, to which methylene blue and maltodextrin were added at concentrations of 0.1% and 10%, respectively. The mixture was shaken on a vortex mixer until the added components (i.e., methylene blue and maltodextrin) were completely dissolved within the W_1_ phase of the nano-emulsion. A sample of cyclohexane (15 mL) containing 2.5% R972 hydrophobized fumed silica particles was acquired in another 50 mL plastic tube (denoted as O_2_ oil phase). This oil phase was sonicated (Branson Digital Sonifier, SFX 550, Emerson, Brookfield, CT, USA) at 50% power for 30 sec to disperse the silica particles. O_1_/W_1_ nano-emulsions were added to the cyclohexane and homogenized at 10,000 rpm for 1 min (T25 digital ULTRA-TURRAX^®^, IKA, Staufen, Germany) to obtain three different O_1_/W_1_/O_2_ emulsions. The only difference among all O_1_/W_1_/O_2_ emulsions was the inner O_1_/W_1_ nano-emulsions, which were stabilized with different proteins (denoted as O_1_/W_1_/O_2_—WPI, O_1_/W_1_/O_2_—SPI, and O_1_/W_1_/O_2_—PPI).

### 2.5. Determination of Drop Size Distribution of Double Emulsions

The double emulsions were analyzed for size and polydispersity through analysis of the images recorded using an optical microscope (Delphi-X observer, Euromex, Arnhem, The Netherlands) under bright-field and florescence modes. The sizes of around 500 double emulsion droplets were measured using ImageJ 1.53k software [9]. The data were used to generate number-based size distribution curves for each double-emulsion variant using the frequency distribution function of GraphPad Prism 9 (GraphPad Software, Inc., Boston, MA, United States). Furthermore, the mean droplet size (D_d_) and polydispersity index (PDI = [standard deviation/mean]^2^) were also calculated for each data set.

### 2.6. Formation of O_1_/W_1_/O_2_/W_2_ Triple Emulsion

The double emulsions were used to produce O_1_/W_1_/O_2_/W_2_ triple emulsions. The aqueous phase (W_2_) for dispersing the double emulsion consisted of distilled water containing 10% maltodextrin along with 0.5% R816 fumed silica particles. The aqueous phase was first sonicated at 50% power for 1 min to disperse the silica before making the triple emulsion. The triple emulsion was prepared by dispersing 5 mL of O_1_/W_1_/O_2_ emulsion in 15 mL of W_2_ phase via homogenization at 6000 rpm for 30 s. After following this procedure, we successfully prepared three distinct triple emulsions (denoted as O_1_/W_1_/O_2_/W_2_—WPI, O_1_/W_1_/O_2_/W_2_—SPI, and O_1_/W_1_/O_2_/W_2_—PPI). The optical microscopy and droplet size analyses of the triple emulsions were carried out as described in Section 2.5.

### 2.7. Formation of Antibubbles

The triple emulsions were quickly frozen in −80 °C ultra-freezer (BINDER GmbH, Tuttlingen, Germany). Subsequently, the frozen triple emulsions were lyophilized using a freeze dryer (LyoAlfa 15, Telstar, Terrassa, Spain) at −80 °C and 0.01 mbar vacuum conditions for 48 h. Afterward, the three antibubble variants (i.e., O_1_/W_1_/A/W_2_—WPI, O_1_/W_1_/A/W_2_—SPI, and O_1_/W_1_/A/W_2_—PPI) were obtained via rehydration of the freeze-dried material (0.1 g) in 10% maltodextrin solution (10 mL). The freeze-drying resulted in the replacement of O_2_ oil phase of the original triple emulsion with an air phase (A) to produce antibubbles. Figure 1 illustrates all the required steps involved in the formation of antibubbles. The optical microscopy and particle size analyses of the antibubbles were carried out as described in Section 2.5. As a benchmark, the triple-emulsion-based antibubbles were compared with standard W_1_/A/W_2_ antibubbles in which the inner aqueous phase was not a nano-emulsion but consisted of just a 10% maltodextrin solution.

### 2.8. Entrapment Efficiency

#### 2.8.1. Entrapment Efficiency of Lipophilic Component

The release of a lipophilic component, i.e., Nile red (NR), from the inner O_1_ phase into the external O_2_ phase during secondary emulsification step was quantified to calculate the entrapment efficiencies for the three O_1_/W_1_/O_2_ double emulsions. However, as cyclohexane is volatile, the use of the double emulsions produced in Section 2.4 could have resulted in an imprecise estimation of entrapment efficiency. Therefore, the entrapment of lipophilic components was estimated by preparing double emulsions using MCT oil in the internal and external oil phases, following the same procedure as described in Section 2.4.

The light-pink-colored double emulsions were subjected to centrifugation (Centrifuge 5804 R, Eppendorf, Hamburg, Germany) at 2000 rpm for 10 min. This resulted in a gentle separation of the oil layer (O_2_) that appeared at the top of each double emulsion. The O_2_ phase was then drawn using a 10 mL syringe, and this portion was then added to a 96-well microplate (MicroWell™, Thermo Scientific™, Loughborough, UK). The absorbance of all samples (Ab_s_) was determined using a UV spectrophotometer (BioTek Epoch 2 microplate spectrophotometer, Agilent, Santa Clara, CA, United States) at a wavelength of 490 nm ($\lambda_{max}$). This absorbance was compared with the absorbance obtained when the entire O_1_ phase had leaked into the O_2_ phase during the secondary emulsification step. This situation was mimicked by mixing O_1_ phase directly into O_2_ phase at the same O_1_/O_2_ ratio as that originally present in O_1_/W_1_/O_2_ double emulsion. Subsequently, the absorbance of this O_1_ and O_2_ mixture (Ab_max_) was determined at the same wavelength, and the entrapment of lipophilic component was calculated as follows:(1)$${EE}_{NR}(\%)=\frac{{NR}_{max}-{NR}_{s}}{{NR}_{max}}\times 100$$
where NR_s_ and NR_max_ are the NR concentrations (mg/mL) corresponding to Ab_s_ and Ab_max_, respectively, which were obtained using a calibration curve obtained with known NR concentrations.

#### 2.8.2. Entrapment Efficiency of Hydrophilic Component

The release of a hydrophilic component, i.e., methylene blue (MB), from the inner W_1_ phase into the outer W_2_ phase was measured for all antibubble variants. Briefly, a small amount of antibubble powder (0.1 g) was extracted and added to 10 mL of water containing 10% maltodextrin. The mixture was allowed to rehydrate for 5 min. After shaking, 2 mL of the solution was collected in a microcentrifuge tube (Expell Secure, CAPP^®^, Nordhausen, Germany) and centrifuged (MiPC 12, MiLab, Dubai, United Arab Emirates) at 3000 rpm for 5 min. The clear solution from the center of the microcentrifuge tube was collected with a 5 mL syringe and added to a 96-well microplate, and absorbance was determined using a UV spectrophotometer at 665 nm. The MB concentration (MB_s_) was determined using a calibration curve obtained from known MB concentrations. The entrapment efficiency of each sample was calculated as follows:(2)$${EE}_{MB}(\%)=\frac{{MB}_{max}-{MB}_{s}}{{MB}_{max}}\times 100$$

${MB}_{s}$ and ${MB}_{max}$ represent MB concentrations (mg/mL) in the sample and the maximum possible concentration of MB that can be released into the outer water phase, W_2_. To calculate the maximum possible release of methylene blue (i.e., MB_max_), the antibubbles must be destroyed to release all the methylene blue in the external water phase. For this purpose, Tween 20 was added to the mixture at a concentration of 10%, followed by centrifugation of the mixture at 15,000 rpm for 5 min. Subsequently, the absorbance was recorded at the same wavelength.

### 2.9. Stability of Antibubbles

The stability of all antibubble variants was analyzed in terms of MB release from the cores over an extended period of time (i.e., 2 weeks). The freeze-dried antibubble powder (0.1 g) was rehydrated in 10 mL of 10% maltodextrin aqueous solution. An aliquot (2 mL) from each antibubbles sample was drawn at days 0, 1, 3, 7, 10, and 14 to quantify the release of MB in the W_2_ phase at each storage interval. The procedure used for the measurement of MB_s_ and MB_max_ was similar to that described in Section 2.8.2. The cumulative release of MB from each antibubble variant was expressed as follows:(3)$$Cumulative\;release\;(\%)=\frac{{MB}_{s}}{{MB}_{max}}\times 100$$

### 2.10. Data Analysis

All the experiments and measurements were carried out in triplicate, and then the mean and standard deviation of each data set were calculated using Microsoft Excel 2019 (Microsoft, Redmond, WA, USA). Subsequently, the data were presented graphically by plotting the mean values using GraphPad Prism 9 (GraphPad Software, Inc., Boston, MA, United States). The standard deviation for each data set was represented using errors bars.

## 3. Results and Discussion

### 3.1. O_1_/W_1_ Primary Emulsions

The soluble protein fraction in W_1_, which was used to stabilize the O_1_/W_1_ primary emulsion and obtained after the separation of the insoluble fraction through centrifugation from 2% solutions (as described in Section 2.2), was found to be 1.97% for WPI, 0.59% for SPI, and 0.36% for PPI. The limited solubility of plant proteins has previously been documented, as reported by [10]. The droplet size distributions of the O_1_/W_1_ primary emulsions stabilized by SNP or by the soluble fractions of WPI, SPI, and PPI are shown in Figure 2. The median diameter of the emulsion droplets was around 110 nm for the soy- and pea-stabilized nano-emulsions and around 90 nm for the whey-stabilized emulsion. The latter may be due to the relatively higher soluble fraction of WPI. The median diameter of the SNP-stabilized primary emulsion was 22.1 µm, which is too large to be incorporated in double and triple emulsions. Therefore, the SNP-stabilized primary emulsion was not investigated further regarding antibubble formation. The droplet size distribution for all protein-stabilized primary nano-emulsions was in the range of 0.01 µm to 1 µm, as also reported previously [11,12].

### 3.2. O_1_/W_1_/O_2_ Double Emulsions

After adding MB (representing the hydrophilic payload) to the continuous phase of the O_1_/W_1_ primary emulsion, the primary emulsion was dispersed in another oil phase (i.e., volatile cyclohexane) to prepare the O_1_/W_1_/O_2_ double emulsion. Kindly note that NR had previously been incorporated into the O_1_ phase as a lipophilic payload. The outer interface of the double emulsions was stabilized by adsorbed hydrophobized R972 silica particles since they were previously proven to yield stable antibubbles [2]. The images of all three double-emulsion variants, i.e., O_1_/W_1_/O_2_—WPI, O_1_/W_1_/O_2_—SPI, and O_1_/W_1_/O_2_—PPI, recorded using an optical microscope and a fluorescence microscope, are shown in Figure 3. The retention of NR within the emulsion droplets is confirmed in the fluorescence images of the double emulsions. The fluorescence images show apparently red-colored water droplets, which must result from the fine dispersion of nano-sized oil droplets containing NR within these water droplets (due to Nile red’s neglectable water solubility [13]). A distinctly dark background surrounding the W_1_ droplets in all of the fluorescence images serves as evidence that the O_2_ phase is devoid of NR, confirming the absence of any leakage of O_1_ into O_2_.

The droplet size distribution of the three double-emulsion variants, i.e., O_1_/W_1_/O_2_—WPI, O_1_/W_1_/O_2_—SPI, and O_1_/W_1_/O_2_—PPI, is also depicted in Figure 3. The mean droplet sizes of the double-emulsion variants were 6.8 µm, 7.1 µm, and 8.1 µm for O_1_/W_1_/O_2_—WPI, O_1_/W_1_/O_2_—SPI, and O_1_/W_1_/O_2_—PPI, respectively. Furthermore, in each of the variants, the maximum droplet size remained under 20 μm.

### 3.3. O_1_/W_1_/O_2_/W_2_ Triple Emulsions

The O_1_/W_1_/O_2_/W_2_ triple emulsions were prepared by dispersing the double emulsions in the external W_2_ phase containing silica R816 nanoparticles. It has been previously demonstrated that R816 silica nanoparticles are capable of creating stable antibubbles [2]. R816 nanoparticles possess an intermediate degree of lipophilicity/hydrophilicity because of having a combination of hexadecyl silane groups and –OH groups on their surfaces, rendering the particles appropriate for stabilizing O/W interfaces. The microscopic images of the three triple-emulsion variants (O_1_/W_1_/O_2_/W_2_—WPI, O_1_/W_1_/O_2_/W_2_—SPI, and O_1_/W_1_/O_2_/W_2_—PPI) are shown in Figure 4. The images indicate that multiple droplets of double emulsion were entrapped within the triple emulsion for all variants. The mean droplet size and droplet size distribution of the three triple-emulsion variants O_1_/W_1_/O_2_/W_2_—WPI, O_1_/W_1_/O_2_/W_2_—SPI, and O_1_/W_1_/O_2_/W_2_—PPI can also be seen in Figure 4. The average size of the O_1_/W_1_/O_2_/W_2_—WPI triple emulsion was typically around 28 µm, whereas for O_1_/W_1_/O_2_/W_2_—SPI and O_1_/W_1_/O_2_/W_2_—PPI, the average droplet size was around 35 µm.

### 3.4. O_1_/W_1_/A/W_2_ Antibubbles

The triple emulsions were freeze-dried, and the resulting dried materials were then rehydrated to produce antibubble variants. As shown in Figure 5, the antibubbles were found to be a good replicate of the parent triple emulsions, as evident from the presence of cores inside the antibubbles. The red arrow represents the outermost interface, i.e., A/W_2_, whereas the blue arrow represents the middle interface, i.e., W_1_/A, of an antibubble structure. The inner O_1_ droplets (i.e., the cores) exist in the W_1_ phase but are invisible due to their extremely small size. However, multiple W_1_ droplets are clearly observable in all the variants of the antibubbles, as shown in Figure 5 and schematically illustrated in Figure 1. This confirms that the incorporation of a nano-emulsion (i.e., the cores) did not adversely impact antibubble formation, irrespective of the protein employed for stabilizing the nano-emulsions (although this was further confirmed through the entrapment of hydrophilic and lipophilic payloads, as discussed in the next section).

The particle size distribution of the three antibubble variants is also depicted in Figure 5, along with the mean particle size (D_p_) and polydispersity index (PDI). The antibubbles loaded with the WPI-stabilized nano-emulsion had the smallest average size, i.e., 22.7 ± 2.1 μm, among the three antibubble variants (similar to the trend observed in the case of parent triple emulsions). The other two variants loaded with SPI- and PPI-stabilized nano-emulsions had average sizes of 24.7 ± 2.9 μm and 25.4 ± 2.6 μm, respectively. However, due to some variation in the average size within the replicates of each antibubble variant, the average size was found to be statistically non-significant (*p* > 0.05) between different antibubble variants. Similarly, a non-significant difference (*p* > 0.05) was found in the polydispersity indices of all the antibubble variants.

### 3.5. Entrapment Efficiency

#### 3.5.1. Entrapment Efficiency of Lipophilic Component

During the production of a secondary (intermediate) emulsion, the O_1_ droplets containing a lipophilic payload (i.e., NR) can potentially coalesce with the O_2_ phase. The leakage of the O_1_ phase into the O_2_ phase is unwanted because we desire maximum retention of the lipophilic drug instead of mixing with cyclohexane, which would otherwise impair antibubble formation. Therefore, we consider the entrapment efficiency of the lipophilic component to be an important parameter, which should be estimated immediately after the formation of O_1_/W_1_/O_2_ double emulsions. As depicted in Figure 6a, we found an entrapment efficiency of at least 80% or higher for all the double-emulsion variants, while the SPI variant had the highest entrapment efficiency, amounting to around 90%. The appreciable entrapment efficiency in the case of all the proteins confirms that the majority of the lipophilic payload remained loaded inside O_1_/W_1_/O_2_ double emulsions. These results are also in agreement with the fluorescence images of the O_1_/W_1_/O_2_ double emulsions (Figure 3), which also verify the absence of NR in the external O_2_ phase (Section 3.2).

#### 3.5.2. Entrapment Efficiency of Hydrophilic Component

The entrapment efficiency of the hydrophilic component (i.e., MB) in the core of the reconstituted antibubbles is shown in Figure 6b. The entrapment percentages differed significantly (*p* < 0.05) among the antibubble variants. Specifically, the SPI and PPI variants exhibited entrapment efficiencies of 76.9% and 81.6%, respectively, while the WPI variant showed a comparatively lower entrapment rate of 70.4%. Previously, in the case of double-emulsion-based antibubbles (i.e., W_1_/A/W_2_) that were stabilized using similar nanoparticles, we found an entrapment efficiency of around 80–85% [2]. Therefore, a comparable entrapment was attained in the case of the triple-emulsion based-antibubbles for which the primary nano-emulsions were stabilized by either PPI or SPI. The difference in the performance of the whey and plant proteins in the formation and stability of the triple-emulsion-based antibubbles is further elaborated in the next section.

### 3.6. Stability of Rehydrated Antibubbles

The stability of double-emulsion-based antibubbles (containing only a hydrophilic payload) has already been established, as reported in our recent study [2]. Likewise, it is imperative to investigate the stability of triple-emulsion-based antibubbles, as they contain an additional structure, i.e., protein-stabilized nano-emulsions inside the cores. The stability of such triple-emulsion-based antibubbles (i.e., O_1_/W_1_/A/W_2_) can be expressed in terms of the release of either the lipophilic (O_1_ phase) or hydrophilic (W_1_ phase) component into the external W_2_ phase. However, it is more logical and intuitive that the hydrophilic component will be released before the lipophilic component (during the potential destabilization of the antibubbles), as the leakage of O_1_ into W_2_ requires more barriers to be crossed compared to the leakage of W_1_ into W_2_. Furthermore, the detection of the lipophilic component in W_2_ would be even more challenging due to its insolubility in W_2_. Therefore, the stability of triple-emulsion-based antibubbles can be judiciously described by considering the release of the hydrophilic component alone.

Figure 7a shows the release of the hydrophilic component from the antibubble variants upon rehydration in an aqueous solution of 10% maltodextrin (maltodextrin was used to avoid destabilization due to osmotic pressure imbalance). A significant variation in the initial (burst) release was found among the antibubble variants, i.e., WPI > SPI > PPI; this was due to the dissolution of non-entrapped dye and corresponds to our earlier discussion on the entrapment efficiency of the hydrophilic component (Section 3.5.2). Despite differences in these initial release values, an identical pattern was observed in all the samples: <10% additional release until day 3, followed by a plateau. The release data were further compared with classical double-emulsion-based antibubbles (i.e., W_1_/A/W_2_), which lack both oil and protein within the inner W_1_ phase (referred to as NP). A non-significant difference (*p* > 0.05) was observed in the release behavior of the triple-emulsion PPI-based antibubbles compared to the double-emulsion-based antibubble variants, suggesting that PPI may be the most suitable candidate (compared to other proteins that were tested) for fabricating stable triple-emulsion-based antibubbles.

Fundamentally, all the proteins tested in this study have demonstrated surface-active properties [14,15,16,17], signifying their ability to displace silica particles from both the liquid–liquid and air–liquid interfaces. This hypothesis was tested by rehydrating W_1_/A/W_2_ antibubbles (i.e., NP) in aqueous media containing different proteins in the W_2_ phase at concentrations similar to the W_1_ phase of the triple-emulsion-based antibubbles (Section 3.1). Upon exposure of the antibubbles to these protein concentrations in W_2_, we observed an accelerated destabilization of the antibubbles during this experiment, as indicated by the higher MB release (Figure 7b). The sensitivity of the blank or NP antibubbles clearly relates to the protein concentrations, which were arranged in the following order: WPI > SPI > PPI. However, these findings (indicating an overall higher release of MB) differ from what is shown in Figure 7a, so we can conclude that no appreciable leakage of protein occurred during the preparation of the triple-emulsion-based antibubbles presented in Figure 7a, even during an extended rehydration.

## 4. Conclusions

The purpose of this study was to develop triple-emulsion-based antibubbles capable of efficiently encapsulating both hydrophilic and lipophilic payloads, a feature not achievable with conventional double-emulsion-based antibubbles. Under specific process conditions, we successfully produced antibubble variants within the 23–25 μm size range, each featuring distinct cores containing hydrophilic and lipophilic payloads. The compartmentalized structures of these triple-emulsion-based antibubbles demonstrated substantial entrapment efficiencies for both lipophilic (ranging from 80% to 90%) and hydrophilic (ranging from 70% to 82%) components. This result suggests that antibubbles can serve as a promising option for fabricating multi-drug delivery systems and have the potential to perform better than other competitive drug delivery systems such as liposomes and multiple emulsions. For instance, liposomes exhibit the disadvantage of low encapsulation efficiencies for hydrophilic compounds, unless an active loading of drug is applied, but this only applies a limited range of drugs [18,19]. Moreover, lipophilic drugs may rapidly diffuse out of the liposome bilayer, leading to reduced drug retention and therapeutic efficacy [20,21,22]. Compared to conventional multiple emulsions, antibubbles offer prolong payload stability, and they also provide triggered release capabilities, as mentioned in the Introduction section of this article. Because of these beneficial properties, the potential of triple-emulsion-based antibubbles as a drug delivery system justifies further in vitro and in vitro investigations conducted, e.g., through the co-delivery of multiple drugs such as daunorubicin (hydrophilic) and paclitaxel (lipophilic). 

## Figures and Tables

**Figure 1 pharmaceutics-15-02757-f001:**
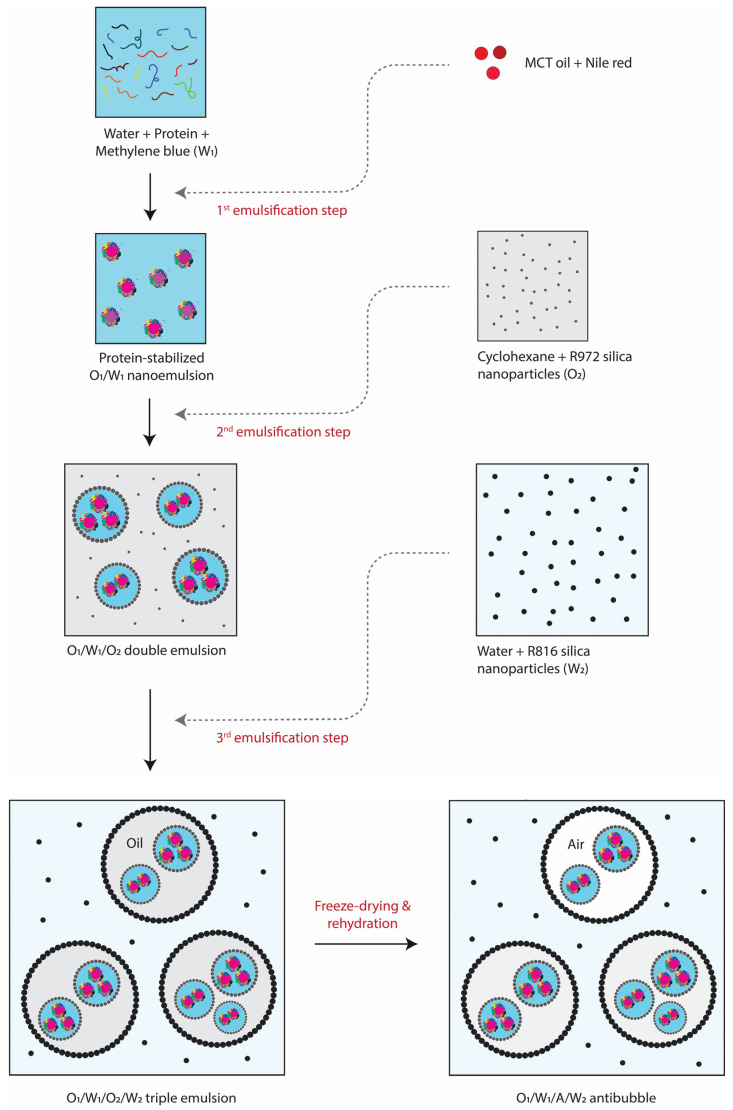
An illustrative representation of the formation process for triple-emulsion-based antibubbles.

**Figure 2 pharmaceutics-15-02757-f002:**
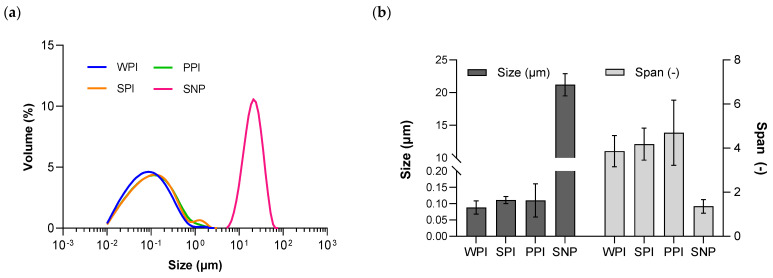
The droplet size distribution (**a**) and the average size (median diameter) and span (**b**) of four O_1_/W_1_ primary emulsions prepared using WPI, SPI, PPI, and SNP.

**Figure 3 pharmaceutics-15-02757-f003:**
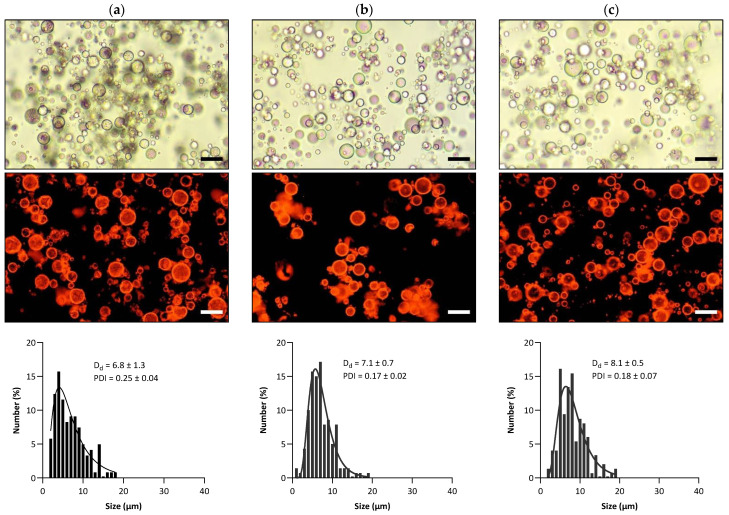
Results of the microscopy (optical and fluorescence) and droplet size analyses of double emulsions: (**a**) O_1_/W_1_/O_2_—WPI, (**b**) O_1_/W_1_/O_2_—SPI, and (**c**) O_1_/W_1_/O_2_—PPI. The scale bars represent a length of 10 μm.

**Figure 4 pharmaceutics-15-02757-f004:**
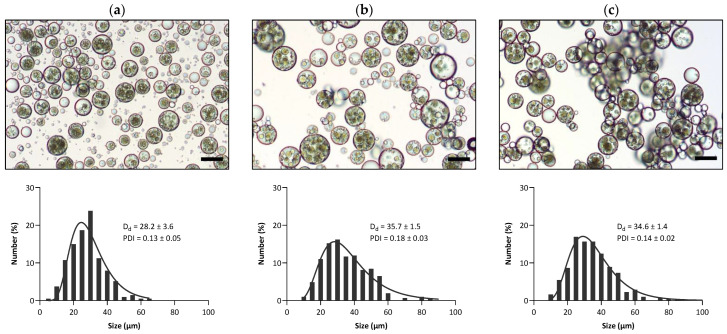
Results of the microscopy and droplet size analyses of triple emulsions: (**a**) O_1_/W_1_/O_2_/W_2_—WPI, (**b**) O_1_/W_1_/O_2_/W_2_—SPI, and (**c**) O_1_/W_1_/O_2_W_2_—PPI. The scale bars represent a length of 30 μm.

**Figure 5 pharmaceutics-15-02757-f005:**
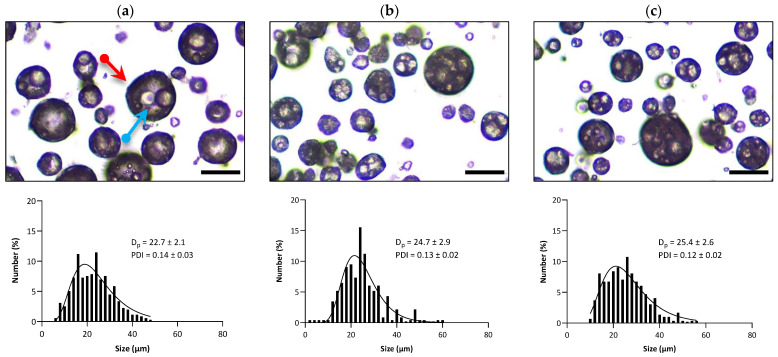
Results of the microscopy and particle size analyses of antibubble (O_1_/W_1_/A/W_2_) variants, for which the primary nano-emulsions were stabilized by various proteins: (**a**) WPI, (**b**) SPI, and (**c**) PPI. The red arrow represents the outermost interface, i.e., A/W_2_, whereas the blue arrow represents the middle interface, i.e., W_1_/A, of an antibubble structure. The scale bars represent a length of 25 μm.

**Figure 6 pharmaceutics-15-02757-f006:**
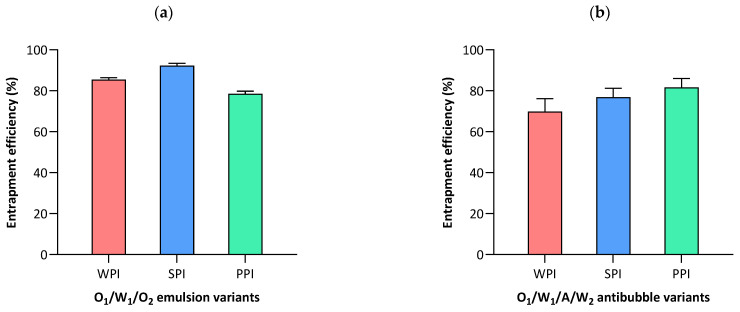
The entrapment efficiency of (**a**) Nile red in the case of O_1_/W_1_/O_2_ double-emulsion variants and (**b**) methylene blue in the case of O_1_/W_1_/A/W_2_ antibubble variants.

**Figure 7 pharmaceutics-15-02757-f007:**
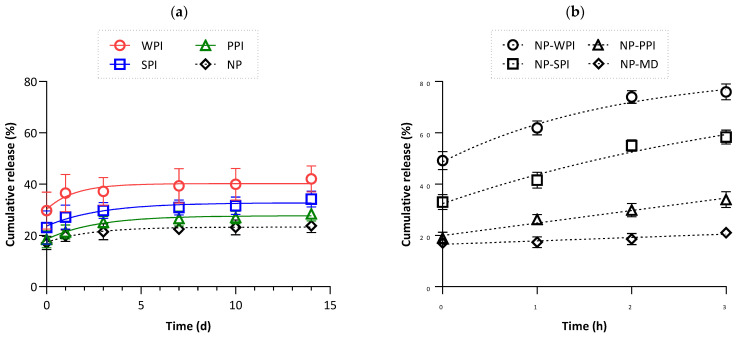
(**a**) The cumulative release of methylene blue from the W_1_ phase of O_1_/W_1_/A/W_2_ (triple-emulsion-based) antibubble variants (WPI, SPI, and PPI) as a function of time; NP refers to double-emulsion-based antibubbles (i.e., W_1_/A/W_2_, without oil and protein in the inner W_1_ phase). (**b**) The cumulative release of methylene blue from NP antibubbles when they were rehydrated in aqueous media containing 10% maltodextrin (MD), either alone or in combination with 2% WPI, SPI, or PPI.

## Data Availability

Data that are contained within the article and the raw data are stored on local servers and are available upon request.

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
