# Peer review of "Triple-Emulsion-Based Antibubbles: A Step Forward in Fabricating Novel Multi-Drug Delivery Systems"

_pharmaceutics, 2023, doi:10.3390/pharmaceutics15122757_

Round 1
Reviewer 1 Report
Comments and Suggestions for Authors
The developed drug delivery system provides a possibiity of biotransportation with high efficiency of both lipophic and hydrophylic substances.
However, the potentials of using such a delivery system were substantiated through using only model systems.
Thus, question appears - how the proposed system will function in vivo, or at least in cell culture, on delivery of water-soluble (ex. doxorubicin) and hydrophobic (ex, paclitaxel) medicines.
The results of such investigation should be presented in the article,
Author Response
As reported in this article, the fabricated antibubbles were able to load model hydrophilic/lipophilic drugs and were stable during the extended rehydration period. We indeed plan to investigate these antibubbles in vitro or in vivo, as suggested by the reviewer, and to report these in a forthcoming paper. Since the other reviewers didn’t see this as an omission, we hope that the editor will allow acceptance of the current manuscript after the revisions according to the comments of the other reviewers.
Reviewer 2 Report
Comments and Suggestions for Authors
The main idea of this work is to develop triple-emulsion-based antibubbles capable of efficiently encapsulating both hydrophilic and lipophilic payloads, a feature not achievable with conventional double-emulsion-based antibubbles. Under specific process conditions, authors successfully produced antibubble variants within the 23-25 μm size range, each featuring distinct cores containing hydrophilic and lipophilic payloads. The compartmentalized structure of these triple-emulsion-based antibubbles demonstrated sub stantial entrapment efficiencies for both lipophilic (ranging from 80% to 90%) and hydrophilic (ranging from 70% to 82%) components. This suggests that antibubbles can serve as a promising option for fabricating multi-drug delivery systems, and have the potential to perform better than other competitive drug delivery systems such as liposomes and multiple emulsions.
Several comments can be made on the article:
Graph 10a needs to be redone by changing the maximum Cumulative release value to 50. This will make the graph more visual
It is not clear from the article which distribution function the authors used to describe the size distribution of bubbles? It is necessary to make appropriate clarifications.
What was the error in determining each of the diameters? How was it calculated? For example, the graphs shown in Figure 10 have a completely different error.
I think it would be interesting to add the dependence of the ratio of the diameter of the inner drop to the diameter of the outer bubble. This will help to understand what kind of payload these means of targeted delivery can carry.
After eliminating these minor corrections, the article can be published
Author Response
… This suggests that antibubbles can serve as a promising option for fabricating multi-drug delivery systems, and have the potential to perform better than other competitive drug delivery systems such as liposomes and multiple emulsions.
Several comments can be made on the article:
Graph 10a needs to be redone by changing the maximum Cumulative release value to 50. This will make the graph more visual.
Authors’ response: For a better visual comparison between Fig. 7a and Fig. 7b (previously Fig. 10), we believe that the scales should be similar. However, the reviewer’s point is also valid. Therefore, in the revised figures we have expanded the y-scale bars to 80 (before it was 100).
It is not clear from the article which distribution function the authors used to describe the size distribution of bubbles? It is necessary to make appropriate clarifications.
Authors’ response: A new text has been added to Section 2.5 to clarify the size distribution for emulsions and antibubbles.
What was the error in determining each of the diameters? How was it calculated? For example, the graphs shown in Figure 10 have a completely different error.
Authors’ response: Section 2.10 has been revised to make this clear.
I think it would be interesting to add the dependence of the ratio of the diameter of the inner drop to the diameter of the outer bubble. This will help to understand what kind of payload these means of targeted delivery can carry.
Authors’ response: This an interesting suggestion, which can be tested in a separate study by keeping the protein type constant and then varying the inner/outer diameter ratio.
After eliminating these minor corrections, the article can be published.
Reviewer 3 Report
Comments and Suggestions for Authors
The authors prepared three triple emulsions based antibubbles (O1/W1/A/W2), and the particle size distribution of the emulsion at each preparation stage was characterized. The entrapment efficiency and stability of rehydrated antibubbles were investigated. However, the study of the manuscript is not deep enough. If the following issues are solved completely, this work might be positively considered for publication.
Comment (1): Why the O1 phase did not sublimate at feeze-drying process?
Comment (2): Why did the three emulsions (WPI, SPI, PPI) differ in the stability of rehydrated antibubbles and the stability of rehydrated antibubbles?
Comment (3): In line 396-397, why is it said that protein particles can replace silica particles at the gas-liquid interface?
Comment (4): Please list the main differences between the two types of silica particles used in the experiment.
Comment (5): The W1 phase in Figure 1 can be made bluer for easier judgment due to the presence of methylene blue.
Comment (6): Please rewrite the conclusion as it was not addressed in any future scope of this work.
Comment (7): What are the innovative points of work? Include your arguments in the manuscript.
Author Response
... If the following issues are solved completely, this work might be positively considered for publication.
Comment (1): Why the O1 phase did not sublimate at freeze-drying process?
Authors’ response: The O1 phase, i.e., the MCT oil in this study, is the solvent for the lipophilic payload which has a low vapor pressure and is therefore not volatile. For obtaining the final antibubble structure, only the O2 phase (i.e., a volatile oil) must sublimate, not the O1 phase.
Comment (2): Why did the three emulsions (WPI, SPI, PPI) differ in the stability of rehydrated antibubbles?
Authors’ response: Overall, all antibubble variants (WPI, SPI, and PPI) were stable during extended rehydration (as seen in Fig. 7a). A difference in stability (i.e. to hold the encapsulated drug) was only observed when the antibubbles were rehydrated in the presence of free protein in the outer aqueous phase (see Fig. 7b). This experiment was done to check possible effects of protein leakage from the inner aqueous phase during the formation stage. To better explain this, we modified the manuscript lines 396 – 408.
Comment (3): In line 396-397, why is it said that protein particles can replace silica particles at the gas-liquid interface?
Authors’ response: This can be explained in two ways: i) The proteins are usually surface active, e.g., as obvious from the results presented in Figure 2. Therefore, they can compete with silica particles at the interface. ii) The proteins can also alter wettability of interfacially adsorbed silica particles, and can lead to destabilization, e.g., as clear from the results presented in Figure 7b.
Comment (4): Please list the main differences between the two types of silica particles used in the experiment.
Authors’ response: New text is provided in Section 2.1 related to the silica particles used in this study.
Comment (5): The W1 phase in Figure 1 can be made bluer for easier judgment due to the presence of methylene blue.
Authors’ response: The W2 phase in Figure 1 is made lighter to create a contrast with the W1 phase.
Comment (6): Please rewrite the conclusion as it was not addressed in any future scope of this work.
Authors’ response: The last part of conclusion is re-written to clarify the future scope of this work (lines 426 - 429).
Comment (7): What are the innovative points of work? Include your arguments in the manuscript.
Authors’ response: The novelty of the work is now highlighted in the Introduction section, lines 63 - 67.
Reviewer 4 Report
Comments and Suggestions for Authors
The current study highlights the innovative concept of antibubbles, a drug delivery system with a unique structure. The study successfully reports the fabrication of triple-emulsion-based antibubbles capable of accommodating both types of payloads, showcasing good entrapment efficiencies and stability over an extended period.
Therefore, I propose that this article shoudl be accepted after the following minor issues are addressed:
difference in the chemistry and size of the AEROSIL® R972, and AEROSIL® R816 should be discussed in more detail in the context of emulsion formation, the Finkle and Bancrof rules.
page 7 line 262 you mention hydrophobized R972, please describe in the experimental section how did you hydrophobize the R972 SNPs.
I find the section 3.4. rather unclear. The O2 was replaced by air, therefore in the paragraph explaining the arrows lines 307-310, in Figure 7a, there should be outer interface i.e. A/W2 and W1/A.
Or, am i getting it wrong?
Also please specify in the caption of Figure 7 if indeed you are looking at the microscope image of the O1/W1/A/W2 Antibubbles?
page 10, lines 310-315 please rephrase the following paragraph as it is not clear...
You didn't provide any fluorescence microscope images with the methylene blue in the water side. Why?
Author Response
… I propose that this article should be accepted after the following minor issues are addressed:
difference in the chemistry and size of the AEROSIL® R972, and AEROSIL® R816 should be discussed in more detail in the context of emulsion formation, the Finkle and Bancroft rules.
Authors’ response: New text has been added in Section 2.1 regarding the application of these silica particles.
page 7 line 262 you mention hydrophobized R972, please describe in the experimental section how did you hydrophobize the R972 SNPs.
Authors’ response: The silica particles were hydrophobized by the supplier, and they were used without any further treatment. This has been made clear now in the Section 2.1.
I find the section 3.4. rather unclear. The O2 was replaced by air, therefore in the paragraph explaining the arrows lines 307-310, in Figure 7a, there should be outer interface i.e. A/W2 and W1/A.
Or, am i getting it wrong?
Authors’ response: The sentence is revised to make it clear to the reader (lines 309 - 317).
Also please specify in the caption of Figure 7 if indeed you are looking at the microscope image of the O1/W1/A/W2 Antibubbles?
Authors’ response: The caption is revised as suggested (Figure 7 is now Figure 5).
page 10, lines 310-315 please rephrase the following paragraph as it is not clear...
Authors’ response: The text is revised to make it clear (lines 309 - 317).
You didn't provide any fluorescence microscope images with the methylene blue in the water side. Why?
Authors’ response: We did fluorescence microscopy of O1/W1/O2, with the aim to visualize any possible leakage of O1 to O2. Unfortunately, we didn’t do the fluorescence microscopy of O1/W1/O2/W2 triple emulsions.
Reviewer 5 Report
Comments and Suggestions for Authors
The work of Cornelus F. van Nostrum and co-authors is devoted to an ongoing topic in materials science - targeted delivery systems. In most cases, modern works describe systems based on genetic vectors that are difficult to obtain, while the presented work uses a potentially industrial method for obtaining targeted delivery systems. This, in my opinion, is a big plus.
The presented work is a self-contained and complete study, and as a reviewer I have only technical comments.
1) Figure 2. Panel A. In my opinion, it is better to give the scale in nanometers.
2) Figure 3. It is better to indicate the scale size in the figures themselves above the scale. It is therefore desirable to add a histogram of bubble size distribution from Figure 4 directly to the microphotograph. this will greatly facilitate the perception of information.
3) Similar to point 2, do for Figures 5 and 6.
4) Similar to point 2, do for Figures 7 and 8.
Otherwise, it turns out that the discussion takes place before the presentation of graphical information, which is not correct.
Author Response
… The presented work is a self-contained and complete study, and as a reviewer I have only technical comments.
1) Figure 2. Panel A. In my opinion, it is better to give the scale in nanometers.
Authors’ response: As majority of the scales are in the μm range and those units are used in the text, we are of the opinion to keep the scale as it is.
2) Figure 3. It is better to indicate the scale size in the figures themselves above the scale. It is therefore desirable to add a histogram of bubble size distribution from Figure 4 directly to the microphotograph. this will greatly facilitate the perception of information.
3) Similar to point 2, do for Figures 5 and 6.
4) Similar to point 2, do for Figures 7 and 8.
Otherwise, it turns out that the discussion takes place before the presentation of graphical information, which is not correct.
Authors’ response: We are thankful to the reviewer for this suggestion, and accordingly we have merged the histograms with the photograph figures. However, the addition of text on top of scale bars would result in more crowding, therefore, we preferred to keep the scale size in the figure caption instead of writing it in all figures. We hope that the reviewer would consider this justification.
Reviewer 6 Report
Comments and Suggestions for Authors
The authors describe the formulation of Triple-Emulsion-based Antibubbles, which is among the novel delivery carriers. The manuscript is well-written and experiments are conducted to validate the system. I have some minor comments to be addressed:
1- Please specify the particle size of the two grades of aerosil used
2- What is the surface charge of the emulsion?
3- The sections Author Contributions, Funding, Data Availability Statement, and Conflicts of Interest are missing
4- Please highlight the novelty of the work in a better way at the end of the introduction section
5- The limitations of the study are not stated
Author Response
… The manuscript is well-written and experiments are conducted to validate the system. I have some minor comments to be addressed:
1- Please specify the particle size of the two grades of aerosil used
Authors’ response: More detail is now provided in Section 2.1 related to the silica particles used in this study.
2- What is the surface charge of the emulsion?
Authors’ response: Unfortunately, we didn’t measure the surface charge of the emulsion.
3- The sections Author Contributions, Funding, Data Availability Statement, and Conflicts of Interest are missing
Authors’ response: The missing sections are now added.
4- Please highlight the novelty of the work in a better way at the end of the introduction section.
Authors’ response: The novelty statement is now highlighted in the Introduction section, lines 63 – 67.
5- The limitations of the study are not stated.
Authors’ response: As reported in this article, the fabricated antibubbles were able to load model hydrophilic/lipophilic drugs and were stable during the extended rehydration period. The limitation could regard to the loading and controlled release of real drugs, which we have planned to investigate in the future. This has been made clear in the conclusion section, lines 426 – 429.
Round 2
Reviewer 3 Report
Comments and Suggestions for Authors
I have no further comments for the paper.